# Computational Repurposing of Mitoxantrone-Related Structures against Monkeypox Virus: A Molecular Docking and 3D Pharmacophore Study

**DOI:** 10.3390/ijms232214287

**Published:** 2022-11-18

**Authors:** Gagan Preet, Emmanuel T. Oluwabusola, Bruce Forbes Milne, Rainer Ebel, Marcel Jaspars

**Affiliations:** 1Marine Biodiscovery Centre, Department of Chemistry, University of Aberdeen, Aberdeen AB24 3UE, Scotland, UK; 2CFisUC, Department of Physics, University of Coimbra, Rua Larga, 3004-516 Coimbra, Portugal

**Keywords:** in silico, monkeypox, drug repurposing, epidemic, poxviruses, pharmacophore, mitoxantrone, molecular docking, zoonotic

## Abstract

Monkeypox is caused by a DNA virus known as the monkeypox virus (MPXV) belonging to the *Orthopoxvirus* genus of the Poxviridae family. Monkeypox is a zoonotic disease where the primary significant hosts are rodents and non-human primates. There is an increasing global incidence with a 2022 outbreak that has spread to Europe in the middle of the COVID-19 pandemic. The new outbreak has novel, previously undiscovered mutations and variants. Currently, the US Food and Drug Administration (FDA) approved poxvirus treatment involving the use of tecovirimat. However, there has otherwise been limited research interest in monkeypox. Mitoxantrone (MXN), an anthracycline derivative, an FDA-approved therapeutic for treating cancer and multiple sclerosis, was previously reported to exhibit antiviral activity against the vaccinia virus and monkeypox virus. In this study, virtual screening, molecular docking analysis, and pharmacophore ligand-based modelling were employed on anthracene structures **(1-13)** closely related to MXN to explore the potential repurposing of multiple compounds from the PubChem library. Four chemical structures **(2)**, **(7)**, **(10)** and **(12)** show a predicted high binding potential to suppress viral replication.

## 1. Introduction

Monkeypox is caused by a DNA virus known as the monkeypox virus (MPXV) belonging to the *Orthopoxvirus* genus of the Poxviridae family, which includes 11 species like smallpox, cowpox, vaccinia, and variola [1], and is endemic to West and Central Africa. Monkeypox is a zoonotic disease with significant primary hosts such as rodents and non-human primates. MPXV can be transmitted to humans by direct contact with rodents (animal bites or direct contact of the animal’s fluids or consumption of these animals), which can then be easily spread by human–human transmission [2]. Human-to-human transmission can occur by close contact with skin lesions or respiratory secretions of an infected individual [3]. In the current outbreak, monkeypox is classified as a sexually transmitted disease (STD), and the smallpox vaccine is an 85% effective preventive measure. [4,5].

To date, monkeypox cases have been transferred to Singapore [6], Britain [7], Israel [8], and America [9] by travelers. The first case of monkeypox was reported in 1970 and diagnosed in the Democratic Republic of Congo (DRC) in Central Africa [2]. Bunge et al. (2022) recorded unsteady fatality rates ranging from 3.6 to 10.6% between two clades of MPXV, the central African clade and the western African clade [2], hypothesizing a growing potential for lethality.

Symptoms of monkeypox are a body temperature between 38.5 °C and 40.5 °C with malaise, a rash and headaches, swelling of the lymph nodes with the presence of complex and deep, well-circumscribed and umbilicated lesions [1]. The incubation time between 7 to 17 days, with the body temperature falling after three days after rash onset [1]. The lesions are swollen, stiff, and painful. Lymphadenopathy, which exists in monkeypox but not smallpox, has been hypothesized to elicit a more robust immune response than smallpox [10]. After the lesions segment, scab formation occurs and eventually these shed off, leading to pale spots [8]. Occasions of sepsis due to lesions have been reported but are generally accepted to be rare [11].

The diagnosis of monkeypox can be made by PCR testing, culturing, immunohistochemistry such as ELISA, or electron microscopy, depending on test availability [5,9]. Viral particles from the case investigated by Erez et al. (2019) showed a particle size of 281 ± 18 nm × 220 ± 17 nm (*n* = 24). Treatment of monkeypox is supported by scientists at the Centers for Disease Control and Prevention (CDC), United States [9]. Because of the similarity between MPXV and variola virus (smallpox), it is believed that anti-smallpox drugs, such as tecovirimat and cidofovir, are also effective against monkeypox [9]. Other drugs have not been widely explored, which leaves an open knowledge gap in monkeypox treatment. Mitoxantrone (MXN), an anthracycline derivative, an FDA-approved therapeutic for the treatment of cancer and multiple sclerosis was previously reported to exhibit antiviral activity against the vaccinia virus [12,13]. Mitoxantrone demonstrated an EC_50_ of 0.8 μM against monkeypox by a group of researchers [14], and to improve its efficacy, MXN was further tested for synergistic activity with cidofovir. Based on the inhibition shown by MXN against MPXV previously, we conducted a computational study of 13 closely related anthracene chemical structures (Figure 1) docked against three protein structures. Molecular docking analysis was performed to estimate the strength of binding between the tested compounds and previously used three essential proteins of poxviruses [15,16]. Based on all the results, a pharmacophore study was also carried out to generate ligand-based and structure-based feature pharmacophores. Additionally, using a in silico approach, multi-epitome-based studies have been conducted to design a vaccine against emerging MPXV [16,17,18].

## 2. Results and Discussion

### 2.1. Determination of Antiviral Activity Using Molecular Docking

We undertook a molecular docking study to predict the antiviral activity of 13 anthracene compounds (Appendix A). MXN was used as a standard as this compound in previous research inhibited MPXV. Three proteins were used in this docking study. Unlike most DNA viruses, poxviruses replicate in the cytoplasm of host cells. They encode enzymes needed for genome replication and transcription, including their own thymidine and thymidylate kinases [19]. The first protein used for docking was the PDB: 2V54 [20], which is of the vaccinia virus (Vacc-TMPK) with functional enzyme *thymidylate kinase*. The second one was the PDB: 4QWO [21] *A42R profilin-like protein* (small actin-binding protein involved in cell development, cytokinesis, membrane trafficking, and cell motility) from monkeypox virus Zaire-96-I-16. The third protein was the vaccinia virus *D13*, PDB: 6BED [22]; poxviruses differ from classical enveloped viruses because their membrane is acquired from cytoplasmic membrane precursors assembled onto a viral protein scaffold formed by the *D13* protein rather than budding through cellular compartments. It was found three decades ago that the antibiotic rifampicin blocks this process and prevents scaffold formation. This interaction is the target for the chemical compound, which prevents A17 binding, explaining the inhibition of viral morphogenesis.

Docking poses were analyzed and compared to the standard MXN. The thirteen structures were subjected to docking analysis (Appendix A) and the specificities of their interaction with these targets, as shown in (Figure 2 and Figure 3), were investigated. All these docked results were evaluated for convergence. The best-docked complexes were obtained based on binding energies and interacting residues. Docking poses were analyzed and compared to the standard MXN. In all three molecular docking studies (Table 1, Figure 3, Figure 4 and Figure 5), compounds **(2)**, **(7)**, **(10)**, and **(12)** were the best lead-docked structures.

The amino positions 5 and 8 of the anthracene structures play a vital role. Therefore, the changes in the substituents at amino positions 5 and 8 lead to a better fit with binding sites. Four chemical structures **(2)**, **(7)**, **(10)** and **(12)** showed predicted high binding potentials. Out of these, structures **(10)** and **(12)** showed the highest fit because of the presence of substituents at amino position 8 containing hydroxyl and amino groups which are not present in the structure of MXN. Whereas in structures **(2)** and **(7)**, the amino position 8 substituents remained the same as MXN but the absence or presence of substituents at amino position 5 made these structures fit less compared to structures **(10)** and **(12)**.

Ligplots in (Figure 3) and Table 2 show that MXN standard was found to be involved in hydrogen bonding as well as hydrophobic interactions with *thymidylate kinase* amino acid residues Ser15(B), Arg93(B), Asp13(B), Glu142(B), Asn37(B), and Thr18(B) with bond distances 3.34 Å, 2.80 Å, 3.08 Å, 2.78 Å, 2.98 Å, and 2.99 Å, respectively. Tyr35(B) was the only amino acid residue showing hydrogen bonding interaction with MXN with a bond distance of 3.09 Å. Lys14(B), Tyr144(B), Glu145(B), Leu53(B), Arg41(B), Tyr101(B), Lys17(B), and Gly16(B) engaged in hydrophobic interactions with MXN.

Compound **(2)** was found to be involved in hydrogen bonding as well as hydrophobic interactions with *thymidylate kinase* amino acid residues Asp92(B) with distance 2.99 Å, Thr18(B) with distances 2.63 Å and 3.30 Å, Lys14(B) with distance 2.81 Å, Glu145(B) with distance 2.75 Å, Glu142(B) with distance 3.09 Å and Asn37(B) with distance 2.95 Å. Lys17(B), Phe38(B), Arg93(B), Leu53(B), Phe68(B), Tyr101(B), Asp13(B), Arg41(B) and Pro39 (B) were involved in hydrophobic interactions.

Compound **(7)** was found to be involved in hydrogen bonding as well as hydrophobic interactions with *thymidylate kinase* amino acid residues Asp13(B) with distances 2.83 Å and 2.97 Å, Arg93(B) with distance 2.82 Å, Ser15(B) with distance 3.30 Å, Asn37(B) with distances 2.90 Å and 3.23 Å, and Arg41(B) with distance 2.92 Å. Gly16(B), Glu145(B), Tyr144(B), Glu142(B), Leu53(B), Lys17(B), Lys14(B), and Thr18(B) were involved in hydrophobic interactions.

Compound **(10)** was found to be involved in hydrogen bonding as well as hydrophobic interactions with *thymidylate kinase* amino acid residues Thr18(B) with distance 2.83 Å, Asp13(B) with distance 2.92 Å, Ser97(B) with distance 3.30 Å, Arg93(B) with distance 2.79 Å, and Ser15(B) with distance 3.27 Å. Gly16(B), Lys17(B), Asn37(B), Asp92(B), Arg41(B), Glu142(B), Phe68(B), Tyr101(B), Tyr144(B), and Lys14(B) were involved in hydrophobic interactions.

Compound **(12)** was found to be involved in hydrogen bonding as well as hydrophobic interactions with *thymidylate kinase* amino acid residues Asp13(B) with distances 2.70 Å, 2.92 Å and 3.01 Å, Arg93(B) with distance 2.18 Å, Glu142(B) with distance 2.83 Å, and Asn37(B) with distance 3.13 Å. Tyr101(B), Leu53(B), Glu145(B), Tyr144(B), Thr18(B), Asp92(B), Arg41(B), Gly16(B), Lys14(B), Ser15(B), and Lys17(B) were involved in hydrophobic interactions.

Ligplots in (Figure 4) and Table 3 showed the MXN standard to be involved in hydrogen bonding as well as hydrophobic interactions with the *A42R profilin* amino acid residues Asn14(A) with distances 2.80 Å and 2.90 Å, Thr120(A) with distances with 3.19 Å and 3.30 Å. Thr126(B), with a distance of 3.20 Å, was the only amino acid residue showing hydrogen bonding interaction with MXN. Ala130(B), Ala129(B), Arg127(A), Asp10(A), Phe17(A), Lys16(A), Asn78(B), Asp116(A), Arg119(A), Asp123(A), Tyr80(B), and Glu77(B) engaged in hydrophobic interactions with MXN.

Compound **(2)** was found to be involved in hydrogen bonding as well as hydrophobic interactions with *A42R profilin* amino acid residues Glu77(B) with a distance of 3.24 Å, Asn14(A) with a distance of 2.89 Å, and Thr120(A) with distance 2.86 Å. Asp123(A), Asp10(A), Lys16(A), Asp116(A), Arg119(A), Asn78(B), and Phe17(A) were involved in hydrophobic interactions.

Compound **(7)** was involved in hydrogen bonding and hydrophobic interactions with *A42R profilin* amino acids residue Asn14(A) with distances 2.86 Å and 2.88 Å. Thr126(B), with a distance of 3.11 Å, was the only amino acid residue showing hydrogen bonding interaction. Ala130(B), Arg129(B), Arg127(A), Asp10(A), Thr120(A), Arg119(A), Lys16(A), Phe17(A), Asp116(A), Asn78(B), Asp123(A), Tyr80(B), and Glu77(B) engaged in hydrophobic interactions.

Compound **(10)** was found to be involved in hydrogen bonding as well as hydrophobic interactions with *A42R profilin* amino acid residues Asn14(A) with distances 2.82 Å and 2.88 Å, and Thr120(A) with distance 2.93 Å. Asn78(B), Asp10(A), Glu77(B), Asp123(A), Arg129(B), His100(B), Arg127(A), Ala130(B), Phe17(A), Arg119(A), Lys16(A), and Asp116(A) were involved in hydrophobic interactions.

Compound **(12)** was involved in hydrogen bonding and hydrophobic interactions with *A42R profilin* amino acid residue Asn14(A) with distances 2.85 Å and 3.06 Å. Ala130(B), Arg129(B), His100(B), Arg119(A), Asn78(B), Asp123(A), Asp116(A), Lys16(A), Thr120(A), Phe17(A), Asp10(A), and Arg127(A) were involved in hydrophobic interactions.

Ligplots in (Figure 5) and Table 4 showing the MXN standard to be involved in hydrogen bonding as well as hydrophobic interactions with *D13 protein* amino acids residues Asn530(A) with distance 3.22 Å, Asn118(A) with distance 3.26 Å, Lys159(A) with distance 2.80 Å, Asn121(A) with distance 2.81 Å, Asn464(A) with distances 2.75 Å and 3.33 Å, and Thr474(A) with distance 2.87 Å. Glu230(A), Thr468(A), Asn117(A), Ser256(A), Glu114(A), Ser254(A), and Pro232(A) were involved in hydrophobic interactions.

Compound **(2)** was found to be involved in hydrogen bonding as well as hydrophobic interactions with *D13 protein* amino acid residues Glu114(A) with distance 3.17 Å, Asn530(A) with distance 2.95 Å, Asn464(A) with distances 2.65 Å and 3.04 Å, and Asn117(A) with distance 2.99 Å. Asn121(A), Ser470(A), Asn118(A), Ser256(A), Val528(A), Ser254(A), Thr476(A), Gly473(A), and Thr468(A) were involved in hydrophobic interactions.

Compound **(7)** was found to be involved in hydrogen bonding as well as hydrophobic interactions with *D13 protein* amino acid residues Ser256(A), Thr474(A), Asn464(A), Asn121(A), Lys159(A), Trp108(A), and Glu230(A) with distances 2.98 Å, 2.87 Å, 3.23 Å, 2.82 Å, 2.76 Å, 2.96 Å, and 2.86 Å, respectively. Glu114(A), Ser254(A), Asn117(A), Thr468(A), Asn118(A), Lys484(A), Pro161(A), Pro232(A), and Asn530(A) engaged in hydrophobic interactions.

Compound **(10)** was found to be involved in hydrogen bonding as well as hydrophobic interactions with *D13 protein* amino acid residues Ser256(A), Asn464(A), Asn121(A), Lys159(A), Glu230(A), and Thr474(A) with distances 3.06 Å, 3.23 Å, 2.91 Å, 2.80 Å, 3.02 Å, and 2.80 Å, respectively. Glu114(A), Asn530(A), Ser254(A), Asn117(A), Thr468(A), Asn118(A), Pro161(A), Lys484(A), Val528(A)and Trp108(A) engaged in hydrophobic interactions.

Compound **(12)** was found to be involved in hydrogen bonding as well as hydrophobic interactions with *D13 protein* amino acid residues Ser256(A), Asn464(A), Asn121(A), Lys159(A), and Asn118(A) with distances 3.05 Å, 2.96 Å, 2.98 Å, 2.56 Å, and 3.13 Å, respectively. Ser254(A), Thr474(A), Asn117(A), Thr468(A), Trp108(A), Pro161(A), Pro232(A), Ile110(A), Lys484(A), and Asn530(A) engaged in hydrophobic interactions. 

### 2.2. Pharmacophore Evaluation

Using the lowest energy conformers of MXN, **(2)**, **(7)**, **(10)**, and **(12)**, a pharmacophore model was generated. The generated pharmacophore showed five key features: hydrogen bond acceptors (HBAs), hydrogen bond donors (HBDs), hydrophobic interactions (H), positive ionizable area (PI) and aromatic ring (AR). The representative 3D and 2D pharmacophoric features of each compound are shown in (Figure 6). Each compound constitutes individual pharmacophoric features, and from these individual characteristic pharmacophores, a merged pharmacophore with common features was generated, as shown in (Figure 7). This common feature pharmacophore model with a score of 0.9350 from ten generated models (Appendix A) showed certain features: eight HBD, six HBAs, two Hs, two AR and two PI.

## 3. Materials and Methods

### 3.1. Origin of Compounds 

MXN-related compounds dataset of 13 compounds were taken from Pubchem [23] database to identify potential structures for future study (Appendix A). **(1)** 1-amino-5,8-dihydroxy-4-[2-(2-hydroxyethylamino)ethylamino]anthracene-9,10-dione; **(2)** 5,8-dihydroxy-4-[2-(2-hydroxyethylamino)ethylamino]-9,10-dioxoanthracen-1-yl]azanium; **(3)** 1,4-dihydroxy-5-[2-(2-hydroxyethylamino)ethylamino]-8-[2-(hydroxymethylamino)ethylamino]anthracene-9,10-dione; **(4)** 1,4-dihydroxy-5,8-bis [2-(2-hydroxyethylamino)ethylamino]-10H-anthracen-9-one; **(5)** 1-[2-(ethylamino)ethylamino]-5,8-dihydroxy-4-[2-(2-hydroxyethylamino)ethylamino]anthracene-9,10-dione; **(6)** 1-(4-aminobutylamino)-5,8-dihydroxy-4-[2-(2-hydroxyethylamino)ethylamino]anthracene-9,10-dione; **(7)** 2-[[5,8-dihydroxy-4-[2-(2-hydroxyethylazaniumyl)ethylamino]-9,10-dioxoanthracen-1-yl]amino]ethyl-(2-hydroxyethyl)azanium; **(8)** 1,4-dihydroxy-5-[2-(2-hydroxyethylamino)ethylamino]-8-[2-(propylamino)ethylamino]anthracene-9,10-dione; **(9)** 1-[2-(2-aminoethylamino)ethylamino]-5,8-dihydroxy-4-[2-(2-hydroxyethylamino)ethylamino]anthracene-9,10-dione; **(10)** 1,4-dihydroxy-5-[2-(2-hydroxyethylamino)ethylamino]-8-(5-hydroxypentylamino)anthracene-9,10-dione; **(11)** 1,4-dihydroxy-5-[2-(2-hydroxyethylamino)ethylamino]-8-[3-(2-hydroxyethylamino)propylamino]anthracene-9,10-dione; **(12)** 1-[2-(2-aminooxyethylamino)ethylamino]-5,8-dihydroxy-4-[2-(2-hydroxyethylamino)ethylamino]anthracene-9,10-dione; **(13)** 1-[2-[2-(2-aminoethylamino)ethylamino]ethylamino]-5,8-dihydroxy-4-[2-(2-hydroxyethylamino)ethylamino]anthracene-9,10-dione.

### 3.2. Molecular Docking

Molecular docking analysis was performed using the Autodock Vina v.1.2.0 (The Scripps Research Institute, La Jolla, CA, USA) docking software [24]. The receptor site was predicted using LigandScout 4.4.8 (Inte: Ligand) Advanced software [25] (evaluation license key: 81809629175371877209), which identified putative binding pockets by creating a grid surface and calculating the buriedness value of each grid point on the surface. The resulting pocket grid consisted of several clusters of grid points, rendered using an iso surface connecting the grid points to each other. The isosurface represented empty space that may be suitable for creating a pocket. All ligands and protein structures were prepared using the tool Dock Prep with default parameters in Chimera 1.16 [26]. The net charges were set to neutral for all ligands. Compounds **(2)** and **(7)** structures from PubChem contained charged nitrogen, and this was neutralized using Dock Prep before molecular docking. The X-ray crystal structure of the first protein used for docking was PDB: 2V54 [20], which is the vaccinia virus with the functional enzyme thymidylate kinase. The second is PDB: 4QWO [21] A42R profilin-like protein from the monkeypox virus Zaire-96-I-16. The third protein, vaccinia virus D13, PDB: 6BED [22], was retrieved from the Protein Data Bank and utilized to perform docking simulations.

The box center and size coordinates for (PDB: 2V54) were 5.6 × 19.2 × −30.6 and 26.2 × 33.2 × 19.9; for (PDB: 4QWO) was 1.2 × 16.7 × 16.5 and 24.7 × 21.1 × 28.6; for (PDB: 6BED) was 106.0 × 100.4 × 14.0 and 27.9 × 38.6 × 96.4 around the active site.

All coordinates used Angstrom units. The following search parameters were used: the number of binding modes was 10, exhaustiveness was 8, and the maximum energy difference was 3 kcal/mol. Results were tested for convergence at exhaustiveness 16 and 24, keeping all the methods mentioned above the same. Chimera 1.16 [26], LigPlot^+^ 2.2 software [27] and Samson by OneAngstrom, 2022, [28] were used for the visualization and calculation of protein–ligand interactions.

### 3.3. 3D Pharmacophore Model Generation

LigandScout (Inte: Ligand) Expert software [25] (evaluation license key: 44427459425915253797) was used to generate a 3D pharmacophore model. Default conformation generation settings were used where a maximum number of conformations was 50, timeout (sec) was 600, RMS threshold was 0.8, energy window was 20.0, maximum pool size was 4000, maximum fragment build time was 30, slave memory was set to −1.0 and number of slave processes was set to 2. Espresso algorithm was used to generate ligand-based pharmacophores. The generated pharmacophore model compatible with the pharmacophore hypothesis was created using the default settings for LigandScout. Relative pharmacophore fit scoring function with the merged feature pharmacophore type and feature tolerance scale factor was set to 1.0 for ligand-based pharmacophore creation. The number of omitted features for the merged pharmacophore was set to 4, partially matching features optional, threshold (%) was set to 10, and a maximum number of resulting pharmacophores was set to 10. The best model was selected from the ten generated models.

## 4. Conclusions

In this computational study, based on inhibition shown by MXN against MPXV by the Altmann research group, we report that the structures **(2)**, **(7)**, **(10)**, and **(12)** may have the capacity to inhibit the monkeypox virus and synergistically act with cidofovir (which was previously reported for MXN by the Altmann research group). The in silico screening of structures displayed interesting interactions with the binding site of previously reported proteins. These compounds gave good binding potentials. These results proposed a common pharmacophore model that could help guide future studies in identifying, selecting, and designing anthracene structures from big compound library or synthetically producing those that could work individually or synergistically with other drugs to treat monkeypox and other poxviruses. Further, additional studies comprising in silico and in vitro studies related to stability testing could effectively improve future monkeypox virus inhibitor designs for screened compounds. The authors also hope that the recent increase in pandemics and epidemics, namely COVID-19 and monkeypox, will focus more on much-needed research on discovering antiviral agents. Zoonotic diseases are an increasing threat to human survival, and monkeypox, among other poxviruses, is an emerging threat for which better pharmaceuticals are essential.

## Figures and Tables

**Figure 1 ijms-23-14287-f001:**
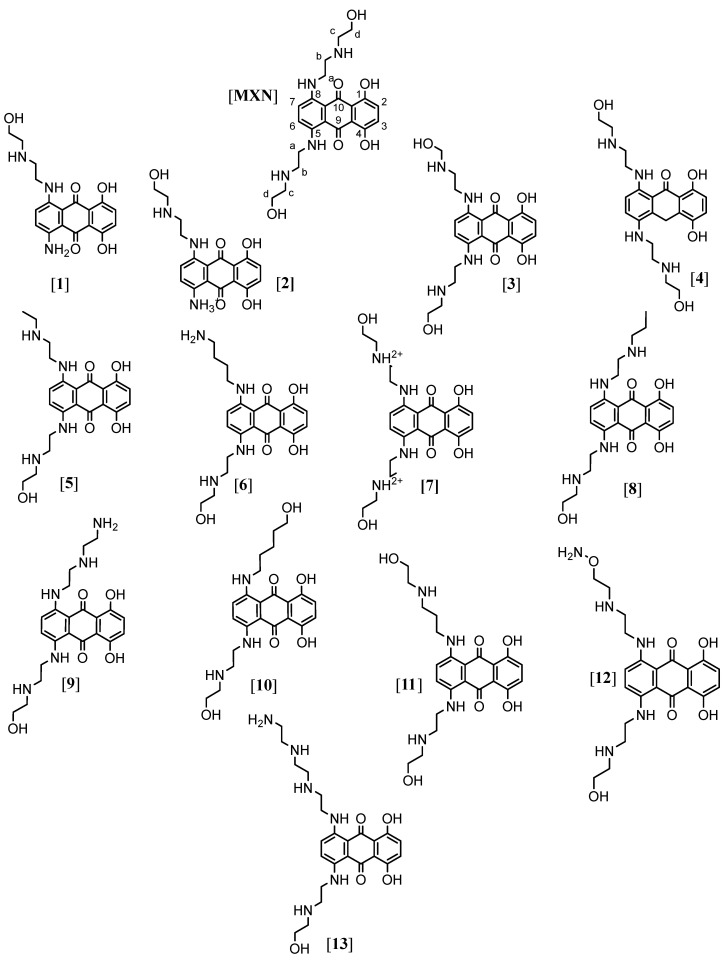
Chemical structures (**1**–**13**) were used in the study.

**Figure 2 ijms-23-14287-f002:**
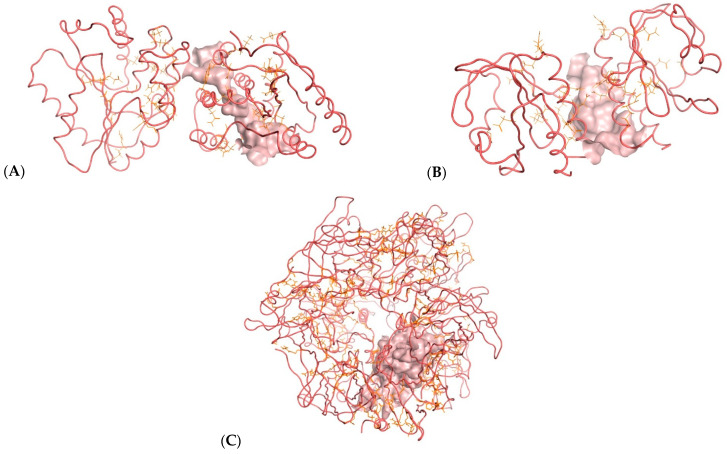
Binding site (solid pink color) of *thymidylate kinase* of the vaccinia virus (**A**); Binding site (solid pink color) of *A42R profilin-like protein* from the monkeypox virus Zaire-96-I-16 (**B**); Binding site (solid pink color) of *D13 protein* of the vaccinia virus (**C**).

**Figure 3 ijms-23-14287-f003:**
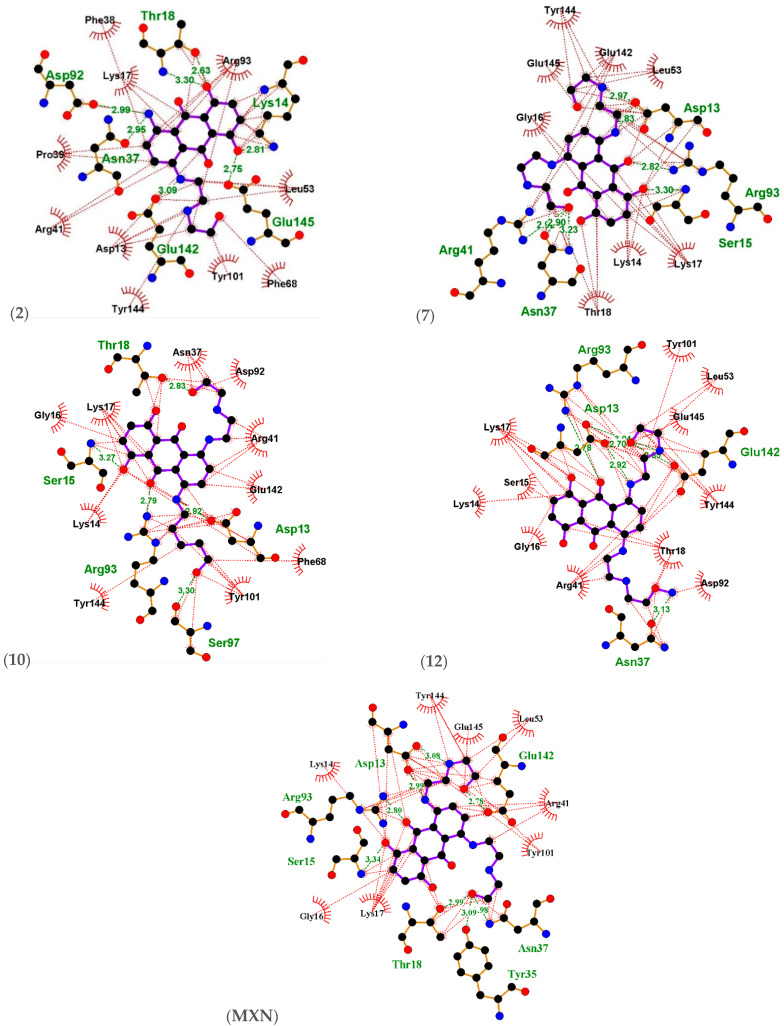
Ligplots showing the interacting residues with the functional enzyme *thymidylate kinase*. Purple lines, anthracene ligand bonds; orange lines, non-ligand bonds; green dotted lines, hydrogen bonds labelled with distances in Å; red dotted lines, hydrophobic interactions; red circles, oxygen atoms; blue circles, nitrogen atoms; black circles, carbon atoms; radial lines, non-ligand residues involved in hydrophobic contact(s).

**Figure 4 ijms-23-14287-f004:**
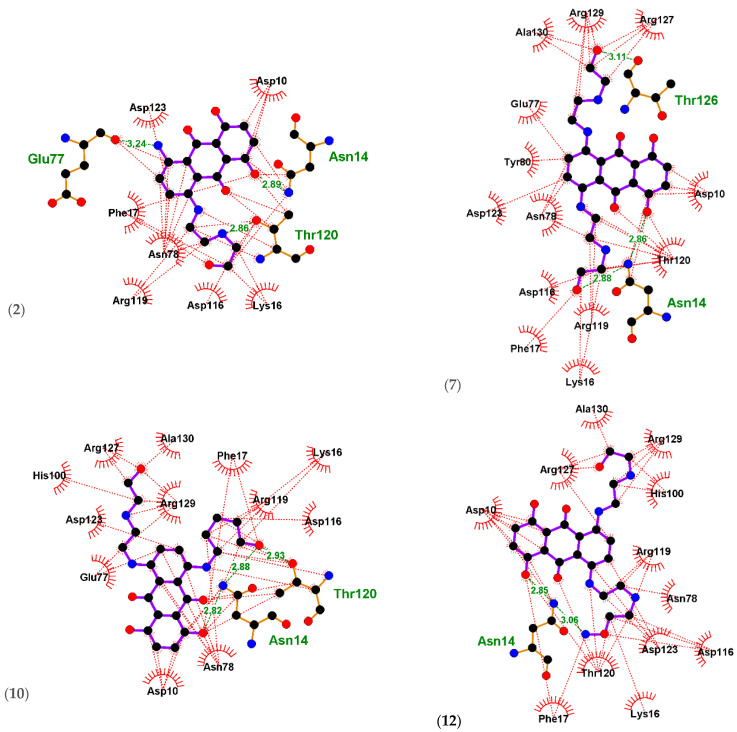
Ligplots showing the interacting residues of the functional enzyme *A42R profilin*. Purple lines, anthracene ligand bonds; orange lines, non-ligand bonds; green dotted lines, hydrogen bonds labelled with distances in Å; red dotted lines, hydrophobic interactions; red circles, oxygen atoms; blue circles, nitrogen atoms; black circles, carbon atoms; radial lines, non-ligand residues involved in hydrophobic contact(s).

**Figure 5 ijms-23-14287-f005:**
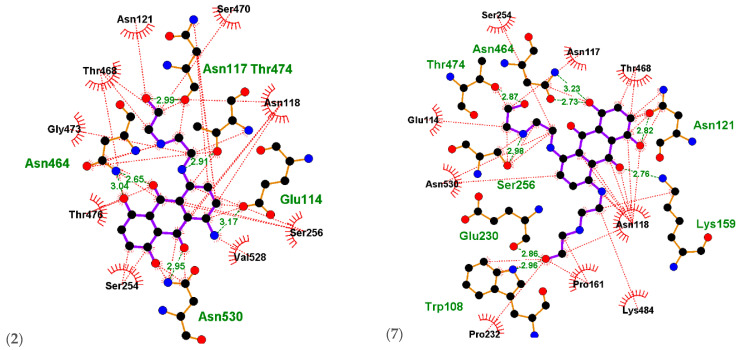
Ligplots showing the interacting residues of the functional enzyme *D13 protein*. Purple lines, anthracene ligand bonds; orange lines, non-ligand bonds; green dotted lines, hydrogen bonds labelled with distances in Å; red dotted lines, hydrophobic interactions; red circles, oxygen atoms; blue circles, nitrogen atoms; black circles, carbon atoms; radial lines, non-ligand residues involved in hydrophobic contact(s).

**Figure 6 ijms-23-14287-f006:**
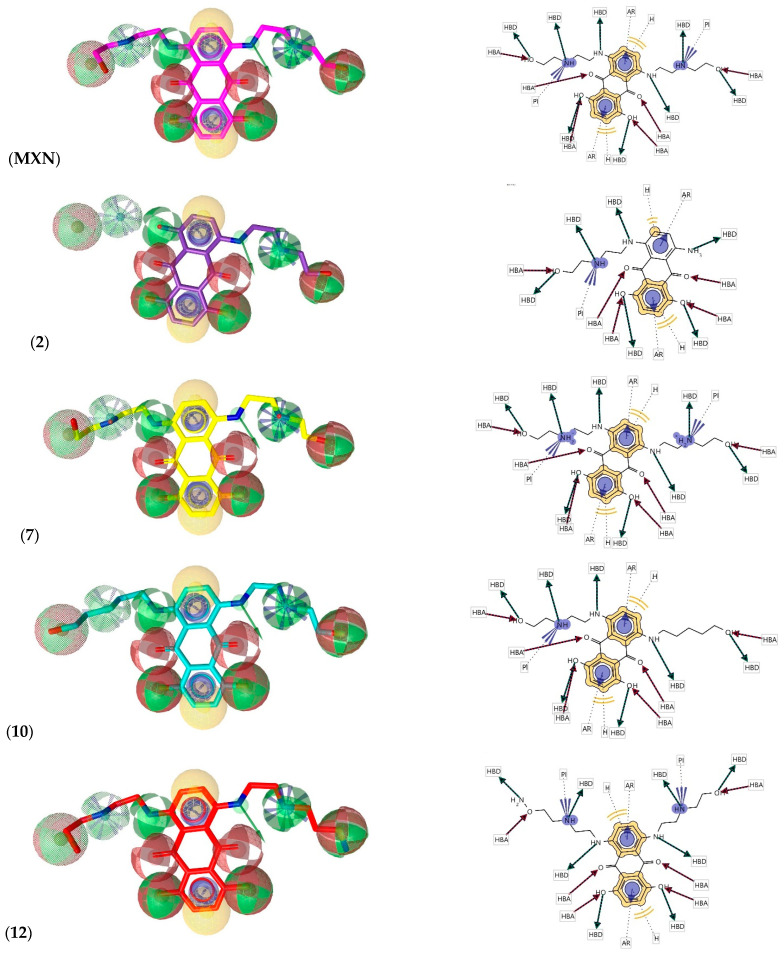
3D and 2D representations of pharmacophoric features of (MXN), **(2)**, **(7)**, **(10)**, and **(12),** used in 3D pharmacophore generation. Red, HBAs; green, HBDs; Yellow, H; Purple, AR; Purple with a cluster, PI.

**Figure 7 ijms-23-14287-f007:**
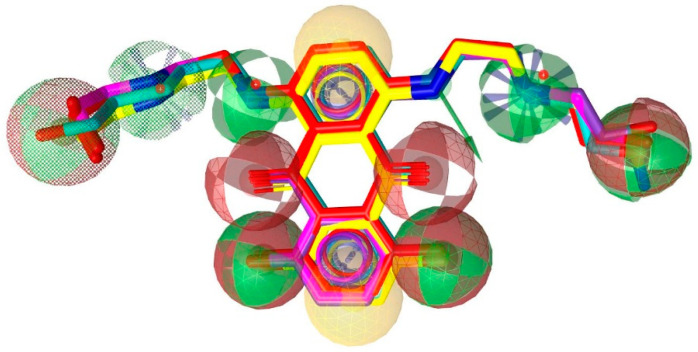
Common feature pharmacophore. Color codes are analogous with Figure 6. Red, HBAs; green, HBDs; Yellow, H; Purple, AR; Purple with a cluster, PI.

**Table 1 ijms-23-14287-t001:** Docking analysis of thirteen structures on three different protein receptors with respect to MXN standard in Blue; best lead compounds in Green.

Compounds (Pubchem ID)	Docking Score (−) (kcal/mol)	Docking Score (−) (kcal/mol)	Docking Score (−) (kcal/mol)
	PDB ID: 2V54	PDB ID: 4QWO	PDB ID: 6BED
	(*Thymidylate Kinase*)	(*A42R Profilin*)	(*D13 Protein*)
MXN (4212) (Standard)	6.9	6.7	7.3
**(1)** (13276605)	7.2	6.6	7.9
**(2)** (153640723)	7.2	6.8	8
**(3)** (44275839)	6.7	6.4	7.6
**(4)** (71044822)	6.9	6.6	7.3
**(5)** (142963046)	6.9	6.1	7.3
**(6)** (44316536)	6.8	6.2	7.5
**(7)** (24848320)	6.9	6.8	7.4
**(8)** (59835539)	7.2	6	7.9
**(9)** (44541200)	7	6.1	7.4
**(10)** (58102019)	7	6.8	7.5
**(11)** (143270488)	7.2	6.6	7.4
**(12)** (145293737)	7	6.8	8.3
**(13)** (44541201)	6.8	6.3	7.5

**Table 2 ijms-23-14287-t002:** Table showing interacting residues of functional enzyme *thymidylate kinase*.

Residues	MXN	Compound (2)	Compound (7)	Compound (10)	Compound (12)
Arg41(B)	**x**	**x**	**x** ** x **	**x**	**x**
Arg93(B)	**x** ** x **	**x**	**x** ** x **	**x** ** x **	**x** ** x **
Asn37(B)	**x** ** x **	**x** **x **	**x** **x **	**x**	**x** **x **
Asp13(B)	**x** **x **	**x** **x **		**x** **x **	**x** **x **
Asp92(B)			**x** **x **	**x**	**x**
Glu142(B)	**x** **x **	**x** **x **	**x**	**x**	**x** **x **
Glu145(B)	**x**	**x** **x **	**x**		**x**
Gly16(B)	**x**		**x**	**x**	**x**
Leu53(B)	**x**	**x**	**x**		**x**
Lys14(B)	**x**	**x** **x **	**x**	**x**	**x**
Lys17(B)	**x**	**x**	**x**	**x**	**x**
Phe38(B)		**x**			
Phe68(B)		**x**		**x**	
Pro39 (B)		**x**			
Ser15(B)	**x** **x **		**x** **x **	**x** **x **	**x**
Ser97(B)				**x** **x **	
Thr18(B)	**x** **x **	**x** **x **	**x**	**x** **x **	**x**
Tyr35(B)	**x**				
Tyr101(B)	**x**	**x**		**x**	**x**
Tyr144(B)	**x**		**x**	**x**	**x**

Hydrogen Bonding—**x**; Hydrophobic—**x**.

**Table 3 ijms-23-14287-t003:** Table showing the interacting residues of *A42R profilin*.

Residues	MXN	Compound (2)	Compound (7)	Compound (10)	Compound (12)
Ala129(B)	**x**		**x**		**x**
Ala130(B)	**x**		**x**	**x**	**x**
Arg119(A)	**x**	**x**	**x**	**x**	**x**
Arg127(A)	**x**		**x**	**x**	**x**
Arg129(B)				**x**	
Asn14(A)	**x** **x **	**x** **x **	**x** **x **	**x** **x **	**x** **x **
Asn78(B)	**x**	**x**	**x**	**x**	**x**
Asp10(A)	**x**	**x**	**x**	**x**	**x**
Asp116(A)	**x**	**x**	**x**	**x**	**x**
Asp123(A)	**x**	**x**	**x**	**x**	**x**
Glu77(B)	**x**	**x** **x **	**x**	**x**	
His100(B)				**x**	**x**
Lys16(A)	**x**	**x**	**x**	**x**	**x**
Phe17(A)	**x**	**x**	**x**	**x**	**x**
Thr120(A)	**x** **x **	**x** **x **	**x**	**x** **x **	**x**
Thr126(B)	**x**		**x**		
Tyr80(B)	**x**		**x**		

Hydrogen Bonding—**x**; Hydrophobic—**x**.

**Table 4 ijms-23-14287-t004:** Table showing the interacting residues of the functional enzyme *D13 protein*.

Residues	MXN	Compound (2)	Compound (7)	Compound (10)	Compound (12)
Asn117(A)	**x**	**x** **x **	**x**	**x**	**x**
Asn118(A)	**x** **x **	**x**	**x**	**x**	**x** **x **
Asn121(A)	**x** **x **	**x**	**x** **x **	**x** **x **	**x** **x **
Asn464(A)	**x** **x **	**x** **x **		**x** **x **	**x** **x **
Asn530(A)	**x** **x **	**x** **x **	**x**	**x**	**x**
Glu114(A)	**x**	**x** **x **	**x**	**x**	
Glu230(A)	**x**		**x** **x **	**x** **x **	
Gly473(A)		**x**			
Ile110(A)					**x**
Lys159(A)	**x** **x **		**x** **x **	**x** **x **	**x** **x **
Lys484(A)			**x**	**x**	**x**
Pro161(A)			**x**	**x**	**x**
Pro232(A)	**x**		**x**		**x**
Ser254(A),	**x**	**x**	**x**	**x**	**x**
Ser256(A)	**x**	**x**	**x** **x **	**x** **x **	**x** **x **
Ser470(A)		**x**			
Thr468(A)	**x**		**x**	**x**	
Thr474(A)	**x** **x **		**x** **x **	**x** **x **	**x**
Thr464(A)		**x**	**x** **x **		
Thr468(A)		**x**			**x**
Trp108(A)			**x** **x **	**x**	**x**
Val528(A)		**x**		**x**	

Hydrogen Bonding—**x**; Hydrophobic—**x**.

## Data Availability

Not applicable.

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
