# Peer review of "Computational Repurposing of Mitoxantrone-Related Structures against Monkeypox Virus: A Molecular Docking and 3D Pharmacophore Study"

_ijms, 2022, doi:10.3390/ijms232214287_

Round 1

Reviewer 1 Report

Authors have described computational study on Mitoxantrone related structures against Monkeypox virus.  However, the manuscript can be improved a lot with rewriting of Introduction, modifying figures quality, incorporating more knowledge in the field of Monkeypox vaccines, improving materials and methods, and more specifically conclusions.

The specific comments, which could help to improve the manuscript, are:

1.      English revision must be performed in grammar (including punctuation).

2.      Abstract: Mention the name or number of all four active compounds.

3.      Introduction section should be improved with some newer information related to new vaccine candidates (Aiman et al., 2022, doi: 10.3389/fimmu.2022.985450;

Abdi et al., 2022, https://doi.org/10.3390/vaccines10091564).

4.      Page 4: line 81-84; “Unlike most DNA viruses ………thymidylate kinase.” Provide a suitable reference for this sentence.

5.      Figures from 3-5 can be improved to highlight the important interacting residues (important interacting residues can be written in bigger font size).

6.      It is recommended to write sequence of amino acid residues in text (page 8-12) as per the sequence of them in table 3-4. Or justify this kind of presentation.

7.      What was the output of Pharmacophore evaluation in this study? Also please elaborate its method section.

8.      Page 16: line 297-299; compound 9 name is repeated.

9.      Authors are suggested to check stability of molecules by MD simulation method, and preferably visualized MD simulation run by VMD.

10.  Authors did not conclude their findings and future directions in an effective way to improve Monkeypox virus inhibitor design. This section (conclusion) of the manuscript needs to be rewritten to remove the redundant text taken from the other sections.

Reviewer 2 Report

Quality of presentation, Ligplots figures and tables are need to be improved  

Reviewer 3 Report

It was handled beautifully. but some corrections are needed.

1) figures 3, 4, and 5 should be made more readable.

2) There are some minor spelling errors in the work done. should be carefully controlled.

Reviewer 4 Report

Preet and co-workers present a computational study to determine the inhibitory potential of 13 anthracene derivative products closely related to mitoxantrone against the Monkeypox Virus. Molecular docking calculations on three related Monkeypox virus enzymes reveal that four compounds have a relevant docking score. A pharmacophore evaluation was performed with the best lead-docked structures with respect to mitoxantrone in order to identify the ligand interaction features.

As the authors point out, zoonotic diseases are an emerging threat and this study supposes one more step to finding new pharmaceutical solutions. However, several deficiencies make this study not acceptable for publication in the International Journal of Molecular Sciences:

1) In the introduction section there is no discussion or citation of previously published computational studies performed on this virus.

2) The discussion of the interactions between every ligand and selected proteins is very repetitive and does not apport any valuable information. It would be more interesting to analyze why the introduction of some substituents to anthracene scaffold leads to a better fit with binding sites instead of only listing the involved residues (which are already summarized in tables 2-4).

3) The quality of figures showing ligand-residues interactions is poor and they are difficult to interpret.

4) Molecular docking studies are not enough to consider a ligand a potential drug. Pharmacokinetics or toxicity endpoints should be explored through in silico tools to ensure the safety and pharmacological activity of the selected molecular structures. Moreover, molecular dynamic simulations can help to determine drug stability under physiological conditions.

5) A reliable pharmacophore evaluation requires more than four presumably active compounds to ensure that identified ligand interaction features can be generalized for every Monkeypox protein.

6) The structure of the text must be revised (for example, lines 69-72 would be better located after line 39).

7) The authors do not provide a Supplementary Information file with the raw data (SMILES or ID of the screened compounds, all predicted binding energies values, etc).

Round 2

Reviewer 1 Report

Authors have justified all comments except one related to MD simulation study. MD simulation study is performed to validate the docking results and to check the stability of molecular interaction.

The manuscript should also be revised to avoid grammatical errors. There are still grammatical errors.

Reviewer 4 Report

The authors have introduced some modifications that improve the quality of the paper. However, two points have not been properly addressed:

2) The discussion of the interactions between every ligand and selected proteins is very repetitive and does not apport any valuable information. It would be more interesting to analyze why the introduction of some substituents to anthracene scaffold leads to a better fit with binding sites instead of only listing in the main text the involved residues (which are already summarized in tables 2-4 and figures).

5) A reliable pharmacophore generation requires more than four presumably active compounds to ensure that identified ligand interaction features can be generalized for every Monkeypox protein.

The previous issues should be addressed and solved before publication.

Round 3

Reviewer 4 Report

The generation of an accurate and generalizable pharmacophore for a specific protein target requires a training set with a great number and variety of active ligands. However, the authors have clarified in the Conclusions section that the generated pharmacophore with four active ligands is only valid for the identification, selection, and design of anthracene structures to treat monkeypox and other poxviruses. Therefore, I consider that all issues have been addressed.